# SAMPLE-AWARE RANDAUGMENT

## ABSTRACT

Automatic data augmentation (AutoDA) improves the generalization of neural networks by filling in the missing data in the target distribution. However, mainstream AutoDA methods suffer from either a time-consuming search process that sets barriers for a wide range of applications, or limited performance due to a lack of dynamic policy adjustments during training. We propose an asymmetric search-free augmentation strategy Sample-aware RandAugment (SRA) that dynamically adjusts the augmentation policy while maintaining a simple implementation. SRA introduces a heuristic score-based module to dynamically evaluate the difficulty of the original training data, which guides the appropriate augmentation independently for each sample. To improve the generalization of the proposed score-based module, SRA adopts an asymmetric augmentation strategy including three steps for two updates: 1) distribution exploration, 2) sample perception, and 3) distribution refinement. In a variety of settings, SRA significantly shrinks the gap between search-based and search-free AutoDA methods. SRA achieves 78.31% ResNet-50 Top-1 accuracy on ImageNet, which is the state-of-the-art among search-free methods. SRA can lead to simpler, more effective, and more practical AutoDA designs for diverse applications in the future.

## 1 INTRODUCTION

Automatic data augmentation (AutoDA) is ubiquitous in training methods and can automatically adjust and explore optimal augmentation policies for various target tasks (Cubuk et al., 2019). It improves the generalization of neural networks for image recognition by filling in the missing data in the target distribution (Lim et al., 2019; Ratner et al., 2017). However, current AutoDA methods generally suffer from prohibitive search time before being applied in training (Cubuk et al., 2019; Tian et al., 2020; Cubuk et al., 2020), or introduce dynamic adjusting policy at the cost of large search overheads (Zhou et al., 2021; Zhang et al., 2019; Kuriyama, 2023). Furthermore, the complicated optimization strategies set barriers for a wide range of applications (Li et al., 2020; Liu et al., 2021a; Hounie et al., 2023), which limit the popularity and applicability of AutoDA to other tasks.

An emerging trend in the field of AutoDA is to design methods that prioritize both simplicity and effectiveness. with the emergency of policy transferring strategies (Cubuk et al., 2019; Lim et al., 2019), AutoDA can be applied without notable performance degradation. In addition, the dramatic search space reduction also allows manual tuning to avoid time-consuming search from scratch (Cubuk et al., 2020). These phenomena boost the development of search-free AutoDA methods (Müller & Hutter, 2021; LingChen et al., 2020; Wightman et al., 2021). The search-free augmentation strategies show great potential by yielding randomly augmented samples of original images. Nevertheless, their capacity to attain the zenith of performance remains limited by their inherent simplicity. For instance, they are unaware of factors such as dataset-specific inclinations towards some transformation operators and deformation levels (Cubuk et al., 2020; Müller & Hutter, 2021).

We carefully summarize two main problems existing in current AutoDA methods: 1) For search-based methods, the complicated time-consuming search process sets barriers for a wide range of applications, and 2) For search-free methods, the suboptimal performance is mainly due to the deficient flexibility to adapt and adjust the policy dynamically during training. These problems inspire us to design a flexible search-free method to dynamically generate effective input for image recognition, without severely increasing the complexity and cost of the training process. To achieve this, we develop a search-free sample-aware AutoDA method named Sample-aware RandAugment (SRA).

Simply put, the core idea of improving the performance of search-free AutoDA methods is to focus more on samples that are more valuable for determining the decision boundaries during training. To achieve this goal, we propose an asymmetric training strategy that splits the original batch into two sub-batches, and augments them with two different policies for exploration and refinement, respectively. During exploration, the target data distribution is explored through random augmentation to improve the representation ability of the target model. During refinement, a heuristic module Magnitude Instructor Score (MIS) based on cosine similarity is proposed to measure the difficulty of each sample in the sub-batch, which further instructs augmentation policy to generate more hard samples that contribute to decision boundaries.

The proposed SRA is evaluated on CIFAR (Krizhevsky et al., 2009) and ImageNet (Russakovsky et al., 2015) benchmarks. Experiments on both convolutional neural networks (CNN) and vision Transformers demonstrate that SRA outperforms current search-free AutoDA methods in a variety of settings, meanwhile achieving competitive or even better performance compared with search-based state-of-the-art methods. We emphasize that SRA achieves 78.31% Top-1 accuracy on ImageNet using ResNet-50 without plenty of tricks, which outperforms the search-free state-of-the-art by 0.24%. In addition, it is also compatible with frameworks such as repeated augmentation (Hoffer et al., 2020) and multi-view contrastive learning (Kurtuluş et al., 2023). As a search-free method, SRA is ready-to-use for a wide range of applications. The contributions are summarized as follows:

- We propose Sample-aware RandAugment (SRA), a search-free sample-aware automatic data augmentation method that shrinks the gap between well-performed yet time-consuming search-based methods and simple yet sub-optimal search-free ones. SRA shows competitive or better performance compared with search-based ones under many settings, demonstrating the potential of search-free heuristic augmentation designs.

- A heuristic module Magnitude Instructor Score (MIS) that dynamically evaluates the difficulty of the original training data is proposed to instruct SRA to generate more samples that contribute to decision boundaries during training. The proposed MIS also provides new insight of sample-awareness in data augmentation to focus on the how to do augmentation to samples rather than simply evaluating the importance of samples.

- We also propose an asymmetric data augmentation strategy that augments samples within one batch through two augmentation policies, aiming at exploring and refining the training data distribution, respectively. The asymmetric strategy provides a new train of thought that the design of hybrid data augmentation is worthwhile for exploration to fully release the power of data augmentation in neural network training.

## 2 RELATED WORK

Automatic data augmentation (AutoDA) has recently appeared and shows significant performance improvement in image recognition. It is generally controlled by a set of policy parameters that determine the deformation, ranges, and sampling probabilities. Compared with human-designed widely applied augmentation methods (Zhong et al., 2020; DeVries & Taylor, 2017; Zhang et al., 2018; Yun et al., 2019), AutoDA methods usually generate images with less synthetic semantics. This technique aims to automatically find a proper augmentation strategy to fill in the missing points in the target data distribution (Lim et al., 2019; Ratner et al., 2017), which is expected to improve the generalization of neural networks.

The development of AutoDA arises from extremely time-consuming search-based methods that require hundreds or thousands of GPU hours to find the optimized policy for the target task even on a proxy that uses a subset or smaller model for policy estimation (Cubuk et al., 2019), which is unrealistic for wide applications. Works afterward try to improve the performance meanwhile reducing the search cost, with techniques such as Bayes optimization (Ho et al., 2019; Lim et al., 2019), weight-sharing strategies (Tian et al., 2020), differentiable learning (Li et al., 2020; Hataya et al., 2020; Liu et al., 2021a), multi-armed bandit algorithm (Lu et al., 2023), or simply expanding the potential augmented image space (Mehta et al., 2022; Zheng et al., 2022). Some AutoDA methods try to dynamically adjust policy during training (Lin et al., 2019; Zhang et al., 2019; Kuriyama, 2023), while requiring repeatedly augmenting the same batch that obviously prolongs training time. In general, search-based AutoDA methods are difficult to simultaneously achieve simplicity, cost-effectiveness, and performance advantages.

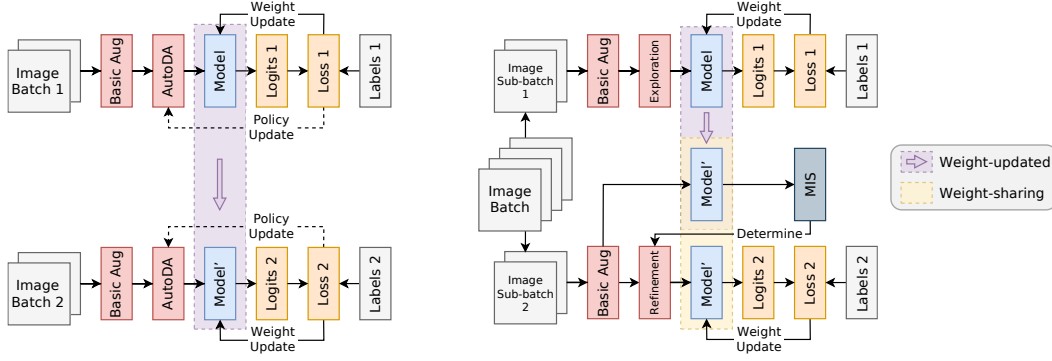

Figure 1: Training pipeline of the proposed Sample-aware RandAugment (SRA) compared with traditional AutoDA. The dotted line illustrates the process in some search-based methods that dynamically adjust policy during training. MIS is the proposed module to evaluate the difficulty of samples.

A rising trend for designing AutoDA is simple yet effective. RandAugment (Cubuk et al., 2020) (RA) was intended to pursue effectiveness while keeping simplicity on the target task to avoid the bias from proxy search. Thanks to the dramatic reduction of the search space, RA is also manually tunable without any search. Meanwhile, methods like TrivialAugment (Müller & Hutter, 2021) and UniformAugment (LingChen et al., 2020) generate augmentation sub-policy through random sampling, which also yields plenty of variants of the original data. In addition, Wightman et al. (2021) hypothesizes magnitudes following a normal distribution in RA, therefore increasing the flexibility of the original RA. These search-free methods only require tuning a few hyperparameters, which is easy to achieve through human priors. The simple heuristic designs show amazing performance that is competitive with many search-based ones, meanwhile, the simplicity makes them more suitable for wide applications. However, the heuristic designs are usually sub-optimal for the target task, therefore search-free methods can hardly achieve the state-of-the-art performance.

Another general problem of the previously mentioned methods is target-aware, neglecting the variations within individual samples in the target task. The idea of customized augmentation boosts the development of sample-aware (Zhou et al., 2021; Lin et al., 2023) or label-aware (Zhao et al., 2022) AutoDA, with which the performance further improves. In particular, MetaAugment (Zhou et al., 2021) uses a policy network for reweighting loss weights to achieve sample awareness, while SelectAugment (Lin et al., 2023) uses two actor-critic structures to determine the suitable samples to apply augmentation. These sample-aware methods mainly focus on the evaluation of importance of different samples rather than the deformation for samples. Meanwhile, they also require complicated optimization strategies to achieve policy learning, which sets barriers to easy implementation.

## 3 SAMPLE-AWARE RANDAUGMENT

### 3.1 REVISIT RANDAUGMENT

RA is a widely applied AutoDA method that boosts the recognition performance of CNNs like ResNets (He et al., 2016) and newly emerging vision Transformers such as DeiT (Touvron et al., 2021) and Swin (Liu et al., 2021b). It only contains two policy parameters that severely reduce the search cost for a direct search on the target task, which determine the level of deformation for all images and the number of augmentation operators to sequentially apply, respectively. The search space reduction that allows direct search without proxy is one of the important contributions of this work. However, the two policy parameters in RA are settled through grid search in the original design, the cost of which is also a heavy burden for wide applications. Thanks to the limited number of policy parameters, these parameters are also tunable through human priors. Meanwhile, the performance of using a transferred policy is competitive with the searched optimal one. Therefore, in practice, RA is widely used without searching, even the policy of which is usually sub-optimal for the target task. Detailed designs of RA, such as the search space including the candidate operators and valid transformation ranges, can refer to Cubuk et al. (2020).

### 3.2 SAMPLE-AWARE RANDAUGMENTATION

We analyze that current mainstream AutoDA methods suffer from either complicated time-consuming search process, or limited performance due to a deficiency of awareness to dynamically adapt and adjust the policy during training. To address the two problems, we propose SRA that

can simultaneously achieve search-free and sample-aware. We add *a heuristic sample perception module* Magnitude Instructor Score (MIS) that uses cosine similarity-based formula to dynamically evaluate the difficulty of the original data during training. Note that focusing only on hard samples may result in a biased represented distribution of the original training data. Therefore, we also introduce *an asymmetric augmentation strategy* that alternatively explores and refines the training data distribution, respectively. The exploration step is designed to adjust the model weights to avoid severe over-fitting on hard samples. The modified pipeline, compared with the traditional one, is shown in Fig. 1. In theory, the strategy is equivalent to training with the alternation of exploration and refinement. To balance the iterations and the number of processed samples of the two policies, we adopt a large batch split strategy in practice, where batch size is twice of the traditional one.

**Step 1: Distribution exploration.** Data augmentation is expected to fill in the missing points of the training data distribution (Lim et al., 2019; Ratner et al., 2017). The augmented image space is usually larger than the original one due to the complex transformations that generate variants of the samples. It is expected to cover more samples in the target data distribution. Therefore, exploring the target distribution with augmented samples is important to avoid models over-fit on the original training data.

Since we have no prior to guide on how to yield beneficial augmented data in the target data distribution, we adopt *a random exploration strategy* to generate variants of the original samples. We sample random operators from the candidate operator set and random magnitudes from a uniform distribution $\mathcal{U}(0, 1)$ to transform the training data. Different augmentation operators may be sequentially applied to form the sub-policy that widens the augmented image space, where $D$ is the number of augmentation operators in one sub-policy, or augmentation depth in the following. The magnitude is sampled independently for each operator within the sub-policy.

**Step 2: Sample Perception.** We propose an intuitive sample-aware augmentation strategy that easy samples require heavier deformations while hard samples less, which is expected to generate plenty of hard samples for determining decision boundaries. Evaluating the difficulty of the original training data is necessary to control the deformation of the augmented images. A heuristic sample-aware perception module to evaluate the difficulty of the original images is proposed, which we call Magnitude Instructor Score (MIS). For convenience, this score is directly applied as the magnitude for the augmentation operators. To be directly applied in data augmentation, the score requires two features: 1) The range of the value should be within $[0, 1]$; and 2) Easier samples should have larger scores, while harder samples smaller.

To satisfy the two demands, we simply choose cosine similarity as the basis of MIS, the original value of which is in range $[-1, 1]$. Here, we evaluate the cosine similarity between the softmax-activated logits, or probabilities of each class, of the original image and the label. Therefore, the value exactly lies in the range $[0, 1]$ due to the non-negative characteristic of probabilities. In addition, it also meets the demand of the second feature for MIS.

In the single-class image recognition task, the labels are one-hot vectors, and the cosine similarity represents the predicted probability of the target class. However, inconsistency exists between classification accuracy and confidence of the prediction on image recognition tasks (Papyan et al., 2020), especially when there are numerous target classes. This brings a negative effect to the cosine similarity-based MIS, where tasks with more classes generally get smaller scores compared with tasks with fewer classes. Therefore, we also introduce a scaling hyperparameter $\gamma$ to normalize MIS for different tasks. Therefore, the final applied MIS in this work is denoted as

$$\text{MIS}_i = \cos(\mathbf{p_i^{ori}}, \mathbf{l_i})^\gamma = (\frac{\mathbf{p_i^{ori}} \cdot \mathbf{l_i}}{\|\mathbf{p_i^{ori}}\|\|\mathbf{l_i}\|})^\gamma, \tag{1}$$

where $\cos$ is cosine similarity function, $\gamma$ is the MIS scaling hyperparameter. $\mathbf{p_i^{ori}}$ and $\mathbf{l}_i$ are the prediction and the label of sample $i$. To adjust the value of $\gamma$ in different tasks, we define a new formula that is denoted as

$$\gamma = \frac{\epsilon}{\log c}, \ \epsilon \geq 0, \tag{2}$$

where $\epsilon$ is the hyperparameter to control the normalization scale of MIS, and $c$ is the number of classes in the target task. The restriction $\epsilon \geq 0$ ensures MIS lies in the range $(0, 1]$. We choose this formula because it yields the same MIS for any $c$ when the predicted probability is uniformly

distributed in each class. With this formula, tasks with more classes require smaller cosine similarity to have the same MIS compared to the ones with fewer classes, which compensates for the difficulty of these tasks to have larger confidence in the target class.

**Step 3: Distribution Refinement.** With the MIS calculated in Step 2, sample-aware data augmentation can be conveniently applied during model training. Since one sub-policy may contain multiple augmentation operators while there is only one MIS for each sample, the calculated MIS is shared among all operators in the same sub-policy as the magnitude. This design is similar to the original RA that shares the same magnitude among different operators in the whole augmentation policy. Apart from the magnitudes used in this step, other procedures are the same as Step 1.

Note that although the MIS module can be applied in more augmentation frameworks that require to settle magnitudes for augmentation operators, the proposed SRA is not simply a combination of MIS and RA. The asymmetric augmentation strategy ensures the effectiveness and generalization of MIS, making the designs of SRA as a whole for wide applications.

## 3.3 TRAIN WITH SRA

**Asymmetrically augment batches.** The training pipeline of SRA is different from most of the previous works which augment each batch in the same way to fill the missing points in the training data distribution. Instead, an asymmetric update strategy is adopted to achieve exploration and refinement through three steps. In detail, for each iteration during training, a large batch containing two small sub-batches with a balanced number of samples is randomly sampled from the training dataset, with the first sub-batch going through Step 1 for exploration while the other Step 2 and 3 for refinement. Thereafter, the loss is calculated and the gradients are propagated backward to update model weights. Therefore, the model weights of Step 2 and 3 are shared while different from Step 1. The process is similar to meta-learning that uses nested update steps to estimate the optimal model weights and meta parameters, while different because the "meta parameters" here are model weights as well. In addition, the training data are not separated into different parts for updating model weights and meta parameters, respectively. The pseudo-code of SRA is in Algorithm 1.

---

**Algorithm 1:** Sample-aware RandAugment (SRA)

**Input:** Image batches $B$ and corresponding labels $y$, training dataset $D_{train}$, augmentation depth $D$, candidate operator set $\mathcal{O}$, uniform sampling $\mathcal{U}$, model $\mathcal{M}$, loss function $\mathcal{L}$, softmax function $\delta$, cosine similarity function $\cos$, the MIS scaling hyperparameter $\gamma$

**Output:** Trained model $\mathcal{M}$

1 **for** $(B, y)$ *in* $D_{train}$ **do**
2    Randomly split $(B, y)$ into $(B_1, y_1)$ and $(B_2, y_2)$
3    $N_1$, $N_2$ are the batchsize of $B_1$, $B_2$, respectively
4    **# Step 1 : Distribution Exploration**
5    **for** $I_i$ *in* $B_1$ **do**
6      Sample $\left\{\mathcal{O}_i^d | 0 < d \le D, \mathcal{O}_i^d \in \mathcal{O}\right\}$
7      $\mathcal{A}_i = \mathcal{O}_i^1 \circ \cdots \circ \mathcal{O}_i^D$
8      $I_i^{exp} = \mathcal{A}_i(I_i, \mathcal{U}(0, 1, \text{size} = (D)))$
9    **end**
10    $B_1' = \left\{I_1^{exp}, \cdots, I_{N_1}^{exp}\right\}$
11    $\mathcal{L}(\mathcal{M}(B_1'), y_1).\text{backward}()$
12    **# Step 2 : Sample Perception**
13    $l_2' = \mathcal{M}(B_2)$
14    $MIS = \cos(\delta(l_2'), y_2)^{\gamma}.\text{repeat}(1, D)$
15    **# Step 3 : Distribution Refinement**
16    **for** $I_j$ *in* $B_2$ **do**
17      Sample $\left\{\mathcal{O}_j^d | 0 < d \le D, \mathcal{O}_j^d \in \mathcal{O}\right\}$
18      $\mathcal{A}_j = \mathcal{O}_j^1 \circ \cdots \circ \mathcal{O}_j^D$
19      $I_j^{ref} = \mathcal{A}_j(I_j, MIS_j)$
20    **end**
21    $B_2' = \left\{I_1^{ref}, \cdots, I_{N_2}^{ref}\right\}$
22    $\mathcal{L}(\mathcal{M}(B_2'), y_2).\text{backward}()$
23 **end**

---

**Definition of the search space.** SRA shares several similar designs with RA, one of which is the search space for the augmentation policy. SRA contains 14 candidate augmentation operators that are the same as RA. It also contains multiple augmentation operators in one sub-policy for augmenting one image. The main difference between the search space of SRA and RA is that the magnitudes of SRA are continuous floating-point numbers rather than discrete deformation levels. The ranges and names of these operators are shown in Table A1 in the Appendix. Although the candidate operators can be specifically selected or use learnable weights for sampling, for simplicity, we hypothesize that operators in the candidate operator set are the same important to contribute to the model training, therefore they have the same probability to be sampled and applied.

| Methods | Search-based | CIFAR-10 | | CIFAR-100 | |
|---|---|---|---|---|---|
| | | WRN-28-10 | SS-26-2x96d | WRN-28-10 | SS-26-2x96d |
| AA (Cubuk et al., 2019) | ✓ | 97.4 | 98.0 | 82.9 | 85.7 |
| FastAA (Lim et al., 2019) | ✓ | 97.3 | 98.0 | 82.8 | 85.4 |
| DDAS (Liu et al., 2021a) | ✓ | 97.3 | 98.0 | 83.4 | 85.0 |
| DeepAA (Zheng et al., 2022) | ✓ | 97.56 | 98.11 | 84.02 | 85.19 |
| LA3 (Zhao et al., 2022) | ✓ | **97.80** | 98.07 | **84.54** | 85.17 |
| BDA (Lu et al., 2023) | ✓ | 97.49 | 98.05 | 83.48 | 85.01 |
| RA (Cubuk et al., 2020) | ✓ / ✗ | 97.3 | 98.0 | 83.3 | - |
| UA (LingChen et al., 2020) | ✗ | 97.33 | 98.10 | 82.82 | 84.99 |
| TA (RA) (Müller & Hutter, 2021) | ✗ | 97.46 | - | 83.54 | - |
| TA (Wide) (Müller & Hutter, 2021) | ✗ | 97.46 | **98.21** | 84.33 | **86.19** |
| **SRA (Ours)** | ✗ | **97.67 ± 0.02** | **98.36 ± 0.08** | **84.64 ± 0.04** | **85.74 ± 0.05** |

Table 1: CIFAR results using CNNs with different structures. We label whether the listed AutoDA methods require a search. **RED**: Best performance. **BLUE**: Second best performance.

# 4 EXPERIMENTS AND ANALYSES

We conduct the experiments to evaluate the performance of SRA on three classical benchmarks: CIFAR-10, CIFAR-100, and ImageNet, and compare it with other mainstream AutoDA methods. The performances of other methods are from their original paper if not specially mentioned. However, we notice that the settings of different methods are not the same. For relatively fair comparisons, we also report the performance of our methods under different settings to be comparable to other methods. Apart from benchmark comparisons, we also show the compatibility of SRA with other augmentation frameworks such as Tied Augment (Kurtuluş et al., 2023). We also compare the performance of SRA with online AutoDA methods that integrate repeated augmentation (Hoffer et al., 2020). The hyperparameter settings of our SRA generally follow previous works, which are shown in Table A2 in the Appendix. All experiments are run for three times, the average performance and standard deviations of which are reported for self-implemented experiments.

## 4.1 CIFAR-10 & CIFAR-100

Following previous works, we evaluated our SRA on two models Wide-ResNet-28-10 (Zagoruyko & Komodakis, 2016) (WRN-28-10) and ShakeShake-26-2x96d (Gastaldi, 2017) (SS-26-2x96d). Performances of Top-1 accuracy (%) of different AutoDA methods are shown in Table 1. For a fair comparison, only methods with similar epochs and tricks during training to our work are listed in the table. Note that we also label whether the method requires a search in the table, where search-free methods are expected to be more convenient for wide applications. We mark RA as both search-based and search-free because it has only a few policy parameters that are easily tuned through human priors. TA has two search spaces (RA and Wide), of which RA space is the same as our SRA, while Wide space has wider ranges of magnitudes for each operator.

As a search-free method, SRA outperforms other search-free methods in many cases, while achieving competitive or even slightly better performance than search-based ones. For SS-26-2x96d on CIFAR-100, SRA is slightly worse than TA (Wide). We analyze this is because CIFAR-100 prefers wider magnitude ranges, especially where ShakeShake views more samples in the wider ranges due to the longer training period (1800 epochs). The experiments demonstrate that our SRA can improve the performance of search-free AutoDA methods under many conditions on CIFAR benchmarks.

## 4.2 IMAGENET

To show the performance of SRA on a larger and more challenging dataset ImageNet that contains 1,000 categories and 1.3 million images, we evaluate SRA with a classical CNN model ResNet-50 (He et al., 2016) and a larger one ResNet-200. We compare both Top-1 and Top-5 accuracy on this dataset with other methods. Both performance of SRA with and without label smoothing (Szegedy et al., 2016) are shown in Table 2. Since RA is one of the main focuses for comparison, while it has fewer training epochs (200 vs. 270) on ImageNet in the original paper, we reproduce its results under our settings for a fair comparison. Note that the current reported state-of-the-art is Augmentation-wise Weight Sharing (AWS) (Tian et al., 2020), where the Top-1/5 performance on ResNet-50 is 79.39% and 94.51%, respectively. However, the detailed settings of this method are not mentioned, meanwhile, the code is not available for public use. Considering these factors, we do not list the performance of it in Table 2.

For comparison on ResNet-50, SRA significantly outperforms all search-free methods, while also achieving better performance than many search-based ones. Even though the performance of SRA

| Label Smoothing | Methods | ResNet-50 | | ResNet-200 | |
|---|---|---|---|---|---|
| | | Top-1 | Top-5 | Top-1 | Top-5 |
| No | AA (Cubuk et al., 2019) | 77.6 | 93.8 | 80.0 | 95.0 |
| | FastAA (Lim et al., 2019) | 77.6 | 93.7 | 80.6 | 95.3 |
| | DDAS (Liu et al., 2021a) | 78.0 | - | 80.5 | - |
| | DeepAA (Zheng et al., 2022) | 78.30 | - | **81.32** | - |
| | BDA (Lu et al., 2023) | 78.12 | 93.87 | 80.14 | 95.09 |
| | RA (Cubuk et al., 2020) | 77.6 | 93.8 | - | - |
| | RA (repro.) | $78.06 \pm 0.01$ | $93.82 \pm 0.02$ | $80.43 \pm 0.13$ | $95.16 \pm 0.02$ |
| | UA (LingChen et al., 2020) | 77.63 | - | 80.4 | - |
| | TA (Wide) (Müller & Hutter, 2021) | 78.07 | 93.92 | - | - |
| | **SRA (Ours)** | $\mathbf{78.31 \pm 0.09}$ | $\mathbf{94.02 \pm 0.03}$ | $81.11 \pm 0.09$ | $\mathbf{95.56 \pm 0.02}$ |
| Yes | LA3 (Zhao et al., 2022) | 78.71 | - | - | - |
| | RA (repro.) | $78.53 \pm 0.04$ | $94.20 \pm 0.01$ | $81.00 \pm 0.04$ | $95.32 \pm 0.02$ |
| | **SRA (Ours)** | $\mathbf{78.83 \pm 0.07}$ | $\mathbf{94.24 \pm 0.03}$ | $\mathbf{81.70 \pm 0.05}$ | $\mathbf{95.79 \pm 0.04}$ |

Table 2: ImageNet results using CNNs with different scales. Top-1 and Top-5 accuracy (%) of different AutoDA methods under two settings that are reported.

is not competitive to the state-of-the-art, we emphasize that AWS is an extremely time-consuming method that requires ~157 GPU hours to search for the augmentation policy on ImageNet, which is not practical for out-of-the-box use. On the contrary, our SRA is search-free and only requires minor modifications to the traditional training pipeline, without any complicated search procedure to achieve a performance improvement. While for comparison on ResNet-200, most search-free methods do not report the results. Our SRA outperforms the reproduced RA under both settings, meanwhile achieving competitive performance over the search-based ones. The results demonstrate that SRA shrinks the gap between the search-free and search-based AutoDA methods, especially on models with more parameters and deeper layers.

To further evaluate the generalization of SRA on different neural architectures, we also compare the performance of SRA using vision Transformer DeiT-Tiny (Touvron et al., 2021) without distillation. We show the original performance of DeiT-Tiny for reference. Note that for both SRA and the reproduced RA, Erasing Zhong et al. (2020), repeated augmentation (Hoffer et al., 2020), Mixup (Zhang et al., 2018), and CutMix (Yun et al., 2019) are not applied.

| Methods | Top-1 | Top-5 |
|---|---|---|
| Original | 72.2 | 91.1 |
| RA | $73.76 \pm 0.07$ | $91.41 \pm 0.06$ |
| RA+mag std. | $73.96 \pm 0.12$ | $91.48 \pm 0.09$ |
| **SRA (Ours)** | $\mathbf{74.05 \pm 0.11}$ | $\mathbf{91.55 \pm 0.16}$ |

Table 3: Accuracy (%) on ImageNet using DeiT-Tiny with patch size 16 and resolution $224 \times 224$. RA results are reproduced under our settings.

The results are shown in Table 3, where RA + mag std. is using the RA with standard deviation proposed by Wightman et al. (2021), which is also used in the original DeiT implementation (Touvron et al., 2021). SRA outperforms the existing DA settings, which further demonstrates the potential of SRA for wide applications on different types of neural networks.

## 4.3 COMBINATION WITH TIED AUGMENT

Tied Augment (Kurtuluş et al., 2023) is a newly proposed augmentation framework that is inspired by contrastive learning. It significantly improves the representation ability of models with the alignment of features and logits of different views. Since Tied Augment requires a combination with other data augmentation, we integrate

| Methods | CIFAR-10 | CIFAR-100 |
|---|---|---|
| Tied-RA | **98.1** | 85.0 |
| Tied-RA (repro.) | $97.89 \pm 0.07$ | $84.80 \pm 0.26$ |
| **Tied-SRA (Ours)** | $98.04 \pm 0.05$ | $\mathbf{85.43 \pm 0.14}$ |

Table 4: Top-1 accuracy (%) on CIFAR using WRN-28-10 and Tied Augment.

SRA with it to evaluate the compatibility and performance, which is denoted as Tied-SRA. For a fair comparison, we also reproduce the Tied Augment with RA (Tied-RA) under the same training settings of SRA. The results are shown in Table 4.

Tied-SRA outperforms reproduced Tied-RA on both CIFAR benchmarks, illustrating the advantage of SRA on aligning different views of the original images. However, we note that the reproduced Tied-RA results are slightly worse than the original ones, which may arise from the lack of detailed configurations of the training hyperparameters.

## 4.4 COMBINATION WITH BATCH AUGMENT

The online AutoDA methods that dynamically adapt augmentation policy during training are usually combined with Batch Augment (Hoffer et al., 2020) (BA) that repeatedly augment the same

batch into different variants, aiming at estimating the loss expectations of the augmented images for learning optimal policy parameters. These methods show superior performance compared with the ones without BA, but the duplication of augmentation requires more time for training. To compare our SRA with online methods, we combine SRA with BA to show the performance. We also display the results of the state-of-the-art search-free method TA (Wide) repeated by 8 times as a reference. Since BA allows faster convergence compared with the training without BA (Hoffer et al., 2020), we reduce the training epochs for our SRA (see Table A3 in the Appendix). The results are shown in Table 5. We label the repeated times after each method, which is denoted as ×4 or ×8.

Interestingly, we find SRA outperforms both online search-based AutoDA methods and search-free method TA on CIFAR. We emphasize that SRA achieves state-of-the-art performance with only 200 epochs on SS-26-2x96d under ×8 settings, which saves ∼3 times training

| Methods | CIFAR-10 | | CIFAR-100 | |
|---|---|---|---|---|
| | WRN | SS | WRN | SS |
| MetaA (×4) (Zhou et al., 2021) | 97.76 | 98.29 | 83.79 | 85.97 |
| AdvAA (×8) (Zhang et al., 2019) | 98.10 | 98.15 | 84.51 | 85.90 |
| LatentA (×8) (Kuriyama, 2023) | 98.16 | - | - | - |
| TA (Wide) (×8) (Müller & Hutter, 2021) | 98.04 | 98.12 | 84.62 | 86.02 |
| **SRA (×8) (Ours)** | **98.34** | **98.38** | **85.23** | **86.65** |

Table 5: Accuracy (%) on CIFAR combined with Batch Augment. WRN: WRN-28-10. SS:SS-26-2x96d.

cost compared with other methods that at least require 600 epochs (TA). The augmented images generated by BA estimate the loss expectations of the augmented images generated by the combined AutoDA method. SRA shows better recognition performance through learning from the loss expectations of the augmented samples compared with previous search-free methods, indicating the augmented image distribution of SRA is closer to the target data distribution. The result demonstrates that SRA indeed improves the generalization of represented classes.

## 4.5 TIME COST EVALUATION

Since Step 2 in SRA requires an extra inference to calculate MIS of a sub-batch compared with the traditional training pipeline, the proposed SRA requires ×0.5 extra forward calculation in total. This extra cost is in practical cheap, because the backpropagation rather than forward inference consumes the majority of the training time. To evaluate the practical extra cost of SRA, we also report both the traditional training time and the training time of SRA on CIFAR-100 using single RTX 3090 and WRN-28-10. We find that SRA takes 105 s/epoch, which is only ∼1.1 times the traditional training pipeline (96 s/epoch for RA). With the estimated training cost, we draw the scatter plot of performance (% accuracy) and total training cost (GPU hours) of other AutoDA methods in Fig. 2. The total cost is evaluated according to the

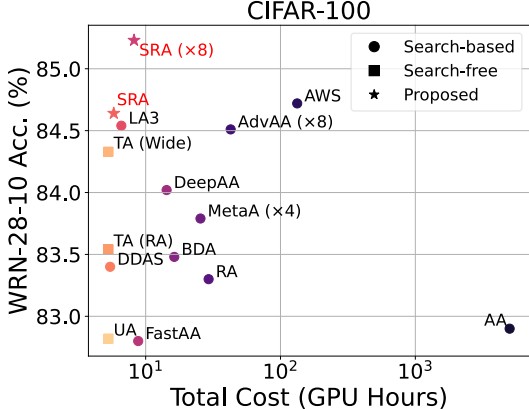

Figure 2: Total search and training cost (GPU hours) of different AutoDA methods.

reported search cost and training epochs in the corresponding papers. Details of how we measure the total cost are listed in Sec. A.3 in the Appendix. As a search-free method, SRA is located at the top-left corner of this figure. It strikes a good balance between performance and time cost.

## 4.6 UNDERSTAND SAMPLE-AWARE RANDAUGMENT

**Ablation studies.** Ablation studies is conducted to measure the contributions of each design in SRA. We split the core designs into five parts: 1) The MIS scaling hyperparameter $\gamma$; 2) Random augmentation for exploring distributions; 3) The distribution exploration process; 4) MIS calculation methods; and 5) Sample perception and distribution refinement. We

| Description | CIFAR-10 | CIFAR-100 |
|---|---|---|
| 1) Remove $\gamma$ | 97.60 | 84.49 |
| 2) Remove random aug in Step 1 | 97.47 | 83.93 |
| 3) Replace Step 1 with Step 2 & 3 | 97.60 | 84.09 |
| 4) Use Euclidean distance for MIS | 97.65 | 84.37 |
| 5) Replace Step 2 & 3 with Step 1 | 97.41 | 83.92 |
| 6) Proposed SRA | **97.67** | **84.64** |

Table 6: Ablation studies of SRA performance (%) using WRN-28-10 on CIFAR.

separately remove or modify each part of these designs, and retrain the models for three times. Results are shown in Table 6.

SRA outperforms all the ablated or modified settings on CIFAR, indicating the effectiveness of the proposed design. We note that both distribution exploration and refinement significantly contribute to the performance, especially on CIFAR-100 which is more challenging. We analyze the effective of SRA is due to the positive regularization effect from hard samples significantly overwhelms the negative overfitting effect. Besides, more formulas to calculate MIS are worth trying, while we find cosine similarity is both intuitive and sufficiently effective for distribution refinement. The results underscore the merits of the asymmetric augmentation strategy in SRA, which may be a catalyst for the advancement of future design of AutoDA methods.

**Represented feature distribution.** We also draw the represented data distribution of SRA in Fig. 3, where augmented data are generated using random sub-policy with corresponding MIS. Features after Global Average Pooling are first reduced to 64 dimensions using Principal Component Analysis, and then shown in 2D t-SNE (Van der Maaten & Hinton, 2008). The figure shows the distributions of the represented features of 10% stratified sampled CIFAR-10 data. The augmented data generally lie at the border of each cluster, indicating the effectiveness of MIS in generating hard samples. Although some augmented samples lie in other clusters due to augment ambiguity (Wei et al., 2020), or are outliers due to over-transformation, they in general benefit the representation of unseen samples.

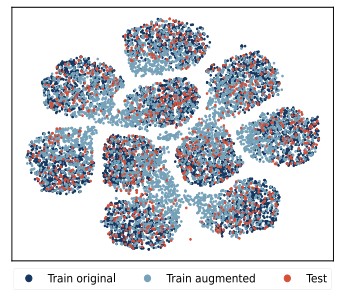 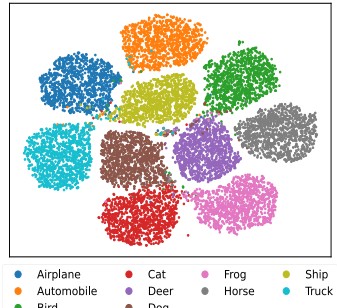

a) Feature Distribution of Each Subset     b) Feature Distribution of Each Class

Figure 3: Feature distributions of CIFAR-10 after SRA augmentation represented through well-trained WRN-28-10.

## 5   LIMITATIONS

Although SRA shrinks the gap between search-based and search-free AutoDA, it has a small search space that limits the performance upper bound. We adopt this design because it is a balance between performance and simplicity. With the selection of augmentation operators through reinforcement learning or gradient optimization, the performance can be further improved. Another existing problem is that mainstream AutoDA methods can hardly avoid the over-transformation, and nor does SRA, due to the lack of further evaluation of the semantics in the augmented samples. Setting more constraints to the augmented data is excepted to alleviate the over-transformation problem. SRA also requires explorations in tasks other than supervised image recognition and downstream task, which requires modifications to the MIS formula and augmentation operators.

## 6   CONCLUSION

In this work, we propose a search-free AutoDA Sample-aware RandAugment to enhance the generalization ability of neural networks. The results demonstrate that heuristic designs can achieve competitive performance to optimized ones, while keeping the simplicity for easy implementation in wide applications. The proposed MIS and asymmetric augmentation strategy may inspire future works to design more novel simple, effective, and practical AutoDA methods, which further contribute to the development of the community.

## 7   REPRODUCIBILITY

The SRA code is accessible in the Supplementary Materials for review, and the camera-ready version will provide the URL. Essential training hyperparameters are documented in Table A2 and A3.

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

# A APPENDIX

## A.1 SEARCH SPACE

| Operator Name | Valid Range |
|---|---|
| ShearX | [-0.3, 0.3] |
| ShearY | [-0.3, 0.3] |
| TranslateX | [-0.45, 0.45] |
| TranslateY | [-0.45, 0.45] |
| Rotate | $[-30°, 30°]$ |
| Brightness | [0.1, 1.9] |
| Color | [0.1, 1.9] |
| Sharpness | [0.1, 1.9] |
| Contrast | [0.1, 1.9] |
| Solarize | [0, 256] |
| Posterize | [0, 4]* |
| Equalize | - |
| AutoContrast | - |
| Identity | - |

Table A1: The candidate 14 operations and corresponding valid ranges in the search space of SRA. *: Implemented using PyTorch, which is identical to [4,8] when using Pillow[1].

Consistent with RA, SRA contains 14 candidate augmentation operators, as in Table A1. And the valid range of the operators are also listed in the table. The main difference between the search space of SRA and RA is that the magnitudes of SRA are continuous floating-point numbers rather than discrete deformation levels.

## A.2 IMPLEMENTATION DETAILS

| | WRN-28-10 | SS-26-2x96d | ResNet-50 | ResNet-200 | DeiT-Tiny |
|---|---|---|---|---|---|
| epochs | 200 | 1800 | 270 | 270 | 300 |
| warmup epochs | 5 | 5 | 5 | 5 | 5 |
| batch size* | 128 * 2 | 128 * 2 | 1024 * 2 | 1024 * 2 | 1024 * 2 |
| learning rate | 0.1 | 0.01 | 0.4 | 0.4 | 1e-3 |
| weight decay | 5e-4 | 1e-3 | 1e-4 | 1e-4 | 0.05 |
| dataset | CIFAR-10/100 | CIFAR-10/100 | ImageNet | ImageNet | ImageNet |
| resolution | 32×32 | 32×32 | 224×224 | 224×224 | 224×224 |
| label smoothing | 0 | 0 | 0 (0.1) | 0 (0.1) | 0.1 |
| dropout | 0 | 0 | 0 | 0 | 0 |
| droppath | - | - | - | - | 0.1 |
| Cutout | 16 | 16 | - | - | - |
| $\epsilon$ | 2 | 2 | 2 | 2 | $\log 1000$ |

Table A2: The hyperparameters for SRA in the comparing experiments with mainstream AutoDA. *: SRA adopts a batch split strategy to update model weights twice with sub-batches respectively, where batch size is required to be twice the comparing methods for a fair comparison. The practical batch size to update model weight is identical to comparing methods regardless of the boundary conditions.

- For experiments on CIFAR, we apply SRA after basic cropping and random horizontal flipping, while before Cutout. For experiments on ImageNet, we only sequentially apply basic cropping, random horizontal flipping, and our SRA. Augmentation depth $D$ for all SRA experiments is set to 2. Other hyperparameters for experiments on CIFAR and ImageNet

[1]https://github.com/python-pillow/Pillow

are listed in Table A2. We also apply a cosine annealing learning rate schedule with a minimum learning rate of 0 for all experiments, which is adjusted after each step for updating model weights.

- For experiments that combine SRA with Tied Augment, the hyperparameter settings are the same as those in CIFAR experiments. The weight to calculate the alignment loss between features of the two views is set to 20 for all experiments, which is the same as the original paper (Kurtuluş et al., 2023).

- For experiments that combine SRA with Batch Augment, the hyperparameters are listed in Table A3.

| | WRN-28-10 | SS-26-2x96d | WRN-28-10 | SS-26-2x96d |
|---|---|---|---|---|
| epochs | 200 | 200 | 35 | 200 |
| warmup epochs | 5 | 5 | 5 | 5 |
| batch size | 128 * 2 | 128 * 2 | 128 * 2 | 128 * 2 |
| learning rate | 0.1 | 0.08 | 0.4 | 0.08 |
| weight decay | 5e-4 | 1e-3 | 5e-4 | 1e-3 |
| dataset | CIFAR-10 | CIFAR-10 | CIFAR-100 | CIFAR-100 |
| repeated times | 8 | 8 | 8 | 8 |

Table A3: The hyperparameters for SRA in the comparing experiments with combined with BA.

All the experiments on CIFAR are conducted on RTX 3090 GPU, while those on ImageNet are conducted on A100. Performances are reported with three different random seeds.

## A.3 TOTAL COST EVALUATION

The total costs in the following are evaluated using WRN-28-10 on CIFAR-100, which is the sum of search overheads and training cost. We use GPU hour as the unit. The basic training time in general is 96 s/epoch, while it is 105 s/epoch for SRA. Therefore, the training cost in general is 96 s/epoch×200 epochs≈5.3 H, while for SRA it is 105 s/epoch×200 epochs≈5.8 H. Methods in the following are listed in the descent of total cost.

**AA:** The reported search overheads is 5000 H, and the training cost is 5.3 H. Thus, in total it costs 5005.3 H.

**AWS:** No directly reported search overheads. Reported search overheads is ×1.5 of OHL (Lin et al., 2019), which is $\frac{1}{60}$ of AA. Therefore, we estimate the training overheads as 5000 H×1.5×$\frac{1}{60}$=125 H. The training epochs are set to 300, therefore training cost is 96 s/epoch×300 epochs=8.0 H. Thus, in total it costs 133.0 H.

**AdvAA (×8):** As an online search-based method, the search overheads is about 0 H. We omit the time for updating policy parameters. The repeated time for one batch during training is 8, therefore the training cost is estimated as 96 s/epoch×200 epochs×8≈42.7 H. Thus, in total it costs 42.7 H.

**RA:** No directly reported search overheads. During the search, 5000 samples are left out for evaluation. Five different policy parameter settings are evaluated. Therefore, we estimate the search overheads as 45000/50000×96 s/epoch×200 epochs×5=24.0 H. The training cost is 5.3 H. Thus, in total it costs 29.3 H.

**MetaA (×4):** As an online search-based method, the search overheads is about 0 H. We omit the time for updating policy parameters. During the search, 1000 samples are left out for evaluation. The reported search epochs is 20, while it takes three times the training time than a standard training scheme (Zhou et al., 2021). The repeated time for one batch during training is 8, therefore we estimate the training cost as (180 epochs+(20 epochs×3×49000/50000))×96 s/epochs×4≈25.5 H. Thus, in total it costs 25.5 H.

**BDA:** The reported search overheads is 11 H, and the training cost is 5.3 H. Thus, in total it costs 16.3 H.

**DeepAA:** The reported search overheads is 9 H. We ignore the influence of deep augmentation to training cost and estimate the training cost the same as in general, which is 5.3 H. Thus, in total it costs 14.3 H.

**FastAA:** The reported search overheads is 3.5 H, and the training cost is 5.3 H. Thus, in total it costs 8.8 H.

**SRA (×8):** As a search-free method, the search overheads is 0 H. SRA combined with BA only takes 35 epochs for training, therefore we estimate the training cost as 35 epochs×105 s/epochs×8≈8.2 H. Thus, in total it costs 8.2 H.

**LA3:** No directly reported search overheads. We run the code provided in the paper and evaluate the search overheads as 1.29 H for Stage 1 and 0.02 H for Stage 2. Therefore, the search overheads is about 1.3 H. The training cost is 5.3 H. Thus, in total it costs 6.6 H.

**SRA:** As a search-free method, the search overheads is 0 H. The training cost is 5.8 H. Thus, in total it costs 5.8 H.

**DDAS:** The reported search overheads is about 0.15 H, and the training cost is 5.3 H. Thus, in total it costs 5.45 H.

**TA (RA), TA (Wide), and UA:** As search-free methods, the search overheads are 0 H for all of the three. Different search space of RA mainly affects the deformation of augmented images, where the time cost for each epoch is almost the same. Therefore, the training cost for TA (Wide) is also 5.3 H. Thus, in total TA (RA), TA (Wide), and UA cost 5.3 H.

## A.4 IMPACT OF NORMALIZATION SCALE $\epsilon$

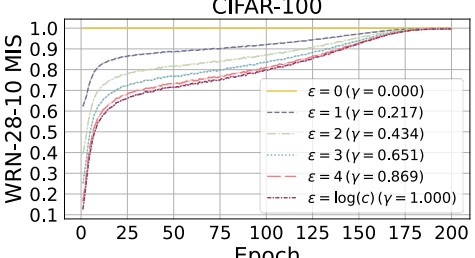

Figure A1: MIS during training on CIFAR-100.

| $\epsilon$ | 0 | 1 | 2 |
|---|---|---|---|
| Accuracy (%) | 84.44 | 84.48 | **84.64** |

| (continue) | 3 | 4 | $\log c$ |
|---|---|---|---|
| | 84.49 | 84.57 | 84.57 |

Table A4: Impact of different values of normalization scale $\epsilon$ on CIFAR-100 performance using WRN-28-10.

We evaluate the impact of normalization factor $\epsilon$ on CIFAR-100 using WRN-28-10, the results of which are shown in Table A4. Meanwhile, we also show how MIS changes during training in Fig. A1. Training with larger $\epsilon$ yields sharper MIS curves, indicating quick changes in the distribution of augmented samples. However, the performance does not monotonically increase with the increase of $\epsilon$. This phenomenon indicates that the adjustment in data distribution should strike a balance between gradual speed and diversity of samples.

We speculate that $\epsilon$ requires careful tuning on the target task to yield the best performance. However, we also find that the MIS without normalization also achieves competitive performance with the best one. Therefore for simplicity, directly using MIS without normalization is a compromise choice when prior information is minor while the training budget is limited, which is adopted in our Transformer experiments. We select $\epsilon = 2$ for all CNNs experiments since it has the best performance.

## A.5 IMPACT OF AUGMENTATION DEPTH $D$

| $D$ | 1 | 2 | 3 | 4 |
|---|---|---|---|---|
| Accuracy (%) | 84.49 | **84.64** | 84.56 | 84.02 |

Table A5: Impact of augmentation depth $D$ on CIFAR-100 using WRN-28-10.

Augmentation depth is an important hyperparameter that determines the scale of the search space. With multiple operators sequentially applied, the diversity of the augmented data increases while

becoming more challenging to learn from. Following previous settings in AA and RA, we mainly select augmentation depth as 2 in our experiments. To evaluate the impact of this hyperparameter on the proposed SRA, we conduct experiments on CIFAR-100 using WRN-28-10. As shown in Table A5, a balance between diversity and the representation ability of the target model is necessary for SRA. Augmentation depth $D = 2$ performs the best, which is consistent with the results in SRA's prototype RA (Cubuk et al., 2020).

### A.6 IMPACT OF AUGMENTATION OPERATORS

We conduct operator ablation experiments on CIFAR-10 using WRN-28-10 in Table A6 to show the sensitivity of SRA to each operator in the search space. For each line, we delete one specific operator from the original 14 operators, and report the performance of SRA on the remaining 13 operators. Each experiment is run for 3 times. As shown, SRA is sensitive to the selection of augmentation operators, which is also the characteristics of previous AutoDA (Cubuk et al., 2020; Li et al., 2020; Zheng et al., 2022). Removing operators like ShearX, TranslateX, Brightness, and Equalize from the configurations of SRA may further improve the performance. However, for simplicity and relatively fair comparisons with previous works, we adopt the candidate operator set chosen in RA.

| Ablated Operator | Accuracy (%) | Operator Gain (%) | Ablated Operator | Accuracy (%) | Operator Gain (%) |
|---|---|---|---|---|---|
| ShearX | 97.78±0.02 | -0.11 | Sharpness | 97.58±0.03 | 0.09 |
| ShearY | 97.56±0.05 | 0.11 | Contrast | 97.65±0.06 | 0.02 |
| TranslateX | 97.70±0.01 | -0.03 | Solarize | 97.60±0.11 | 0.07 |
| TranslateY | 97.60±0.07 | 0.07 | Posterize | 97.51±0.05 | 0.16 |
| Rotate | 97.58±0.05 | 0.09 | Equalize | 97.74±0.01 | -0.07 |
| Brightness | 97.74±0.06 | -0.07 | Autocontrast | 97.59±0.02 | 0.08 |
| Color | 97.59±0.08 | 0.08 | Identity | 97.61±0.04 | 0.06 |

Table A6: Impact of augmentation operators on CIFAR-10 using WRN-28-10. Operators that show negative effect are shown in red.

A prospective way to select candidate operators that are beneficial for training is learning the importance of each operator. Through optimization strategies to select beneficial operators, SRA may be further improved. However, the most important characteristics of SRA is its simplicity for wide application. The optimization of operator selection, which generally uses reinforcement learning, evolutionary algorithms, or gradient optimization, will undoubtedly increase the complexity of SRA, which sets barriers for tuning the proper hyperparameters and realize the implementation in other tasks.

### A.7 ADVANTAGES ON GENERALIZATION TO NEW TASKS

To demonstrate the generalization ability of SRA, we also conduct experiments on a more fine-grained image recognition benchmark Food101 (Bossard et al., 2014). The dataset contains food of 101 categories, with 750 images for training and 250 for validation per class. We treat this benchmark as a new task for exploration, and adopt training configurations of ResNet-50 with label smoothing in Table A2, except no normalization factor $\gamma$ is applied in SRA. We compare the performance of ResNet-50 with different augmentation settings (basic augmentation, RA with 2 operators applied for each sample and value of magnitude 9, and SRA without $\gamma$) on the transferred configurations. Top-1 accuracy on the dats are shown in Table A7, in which all experiments are evaluated on 3 runs.

| Model & Aug Settings | ResNet-50 | | |
|---|---|---|---|
| | Basic | RA(2, 9) | SRA (w/o $\gamma$) |
| Accuracy (%) | 83.18±0.06 | 85.97±0.03 | **87.04±0.02** |

Table A7: The Top-1 accuracy on Food101 benchmark. All models uses the same training hyperparameter settings as in ImageNet experiments (with label smoothing). Evaluated on three runs.

As shown, SRA significantly outperforms the models under the settings of basic augmentation and RA(2,9), indicating a better generalization ability in new tasks. The results also show the potential of SRA to shorten the adaption time and cost for AutoDA method in new applications, which is valuable for the community.

## A.8    LOSS VISUALIZATION

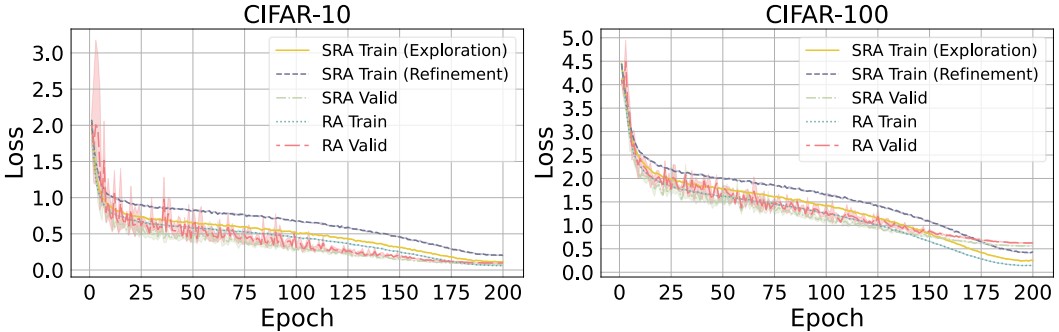

Figure A2: The average loss curves for RA and SRA on CIFAR-10/100 using WRN-28-10 on three runs. The ranges indicate the maximum and minimum loss on each epoch.

We also draw the loss curves of SRA and reproduced RA on CIFAR-10/100 to show how SRA affects the learning process. As shown in Fig. A2, SRA has larger training losses while smaller test losses compared with RA, indicating it generalized better to unknown data. The distribution refinement step shows larger losses compared with the distribution exploration step, which also proves the effectiveness of the distribution refinement step in focusing on hard samples. However, the increase in training loss should be emphasized and accentuated, which may arouse slow convergence that decreases the performance under the same training budget. Meanwhile, it may also increase the difficulty for the target model to represent different classes properly.

