# OpenReview forum: "Sample-aware RandAugment"
_ICLR.cc/2024/Conference — Submitted to ICLR 2024_

### Official Review · Reviewer_dVoG · 2023-10-30

**Soundness:** 3 good
**Presentation:** 3 good
**Contribution:** 2 fair
**Rating:** 5
**Confidence:** 4

**Summary:**

This paper focuses on the task of Automatic Data Augmentation (AutoDA). To achieve simplicity and effectiveness, this paper proposes a method named Sample-aware RandAugment, which dynamically adjusts the magnitude of augmentation operators according to the Magnitude Instructor Score (MIS). MIS shows the consistency between the prediction and the label, which is a heuristic metric to measure the difficulty of samples. Three steps will be adopted during training, including Distribution Exploration (adopting Rand Augmentation with uniformly sampled magnitudes), Sample Perception (measuring MIS), and Distribution Refinement (adopting Rand Augmentation with MIS). The authors conduct experiments on both CIFAR and ImageNet with different network architectures and show performance improvements compared to SOTA AutoDA methods.

**Strengths:**

1. The writing and presentation of this paper is good. The idea is very strightforward and easy to follow.
2. The authors conduct experiments on both CIFAR and ImageNet and adopted the proposed methods with several orthogonal methods to show the effitiveness.

**Weaknesses:**

1. The improvement between the proposed method and RandAugment is marginal. For example, there is only a 0.2% improvement between the SRA and the reproduced RA with ResNet-50 and DeIT in ImageNet experiments.
2. The proposed method highly depends on RandAugment. I consider the most important contribution of this paper to be the strategy of using MIS to adjust the magnitude of each sample. What about adopting the proposed strategy with other augmentation methods? For example, randomly choosing augmentation operators from a search space, and comparing the performances with and without the proposed method.
3. What is the purpose of Step 1? Is it necessary to split the batch into two splits? (1) What about doing step 1 on the first batch and then doing steps 2 and 3 on the second batch? (2) What about removing step 1?

**Questions:**

Please refer to the weaknesses, expecially the second and third weakness.

---

> ### Author Response · Authors · 2023-11-16
> **Response to reviewer dVoG**
>
> We sincerely appreciate your thoughtful comments, efforts, and time. We response to all your concerns as follows:
>
> > **The improvement between the proposed method and RandAugment is marginal. For example, there is only a 0.2% improvement between the SRA and the reproduced RA with ResNet-50 and DeIT in ImageNet experiments.**
>
> Thank you for your comments. Unfortunately, we find that the improvements among AutoDA methods on ImageNet are almost marginal. We emphasize that one of the most important advantages of SRA compared with RA is that SRA doesn’t require manual tuning of hyperparameters if not considering the normalization factor $\gamma$ for calculating MIS. On the contrary, the hyperparameters of RA are found on the basis of previous large search costs on ImageNet. Although the hyperparameters of RA can be manually selected, the cost for adapting it to new tasks apart from basic benchmarks may increase.
>
> To show the advantage of SRA on the adaptation to new tasks, we conduct experiments on Food101 [1], a fine-grained dataset containing foods of 101 classes, with 101,000 images for training and evaluation. We directly use the hyperparameter settings with label smoothing on ImageNet to train ResNet-50 for recognition. We compare the performance of basic augmentation, RA using the transferred strategy from ImageNet (N=2 and M=9, denoted as RA(2, 9)), and SRA without normalization factor $\gamma$ in the following. Interestingly, SRA shows 1.07% accuracy improvement compared with RA, indicating a better generalization ability of it to new tasks. It also avoids the time-consuming search on the whole training dataset as implemented in original RA. ***SRA shows a better generalization ability than RA, which is expected to reduce the adaptation time of AutoDA to new tasks.*** We also emphasize that SRA without $\gamma$ is policy parameter-free when the search space is settled, which makes it easy to be applied as well. Therefore, we believe the application of our SRA as an alternative to RA is worthwhile for wide application.
>
> | Model        | ResNet-50      | ResNet-50      | ResNet-50          |
> | ------------ | -------------- | -------------- | ------------------ |
> | Aug Settings | Basic          | RA (2, 9)      | SRA (no $\gamma$)  |
> | Accuracy (%) | 83.18$\pm$0.06 | 85.97$\pm$0.03 | **87.04$\pm$0.02** |
>
> The new experiment is added in the Appendix, and the discussions about the advantages of SRA over RA is also stressed in the Discussion.
>
> [1]*Bossard L, Guillaumin M, Van Gool L. Food-101–mining discriminative components with random forests[C]//Computer Vision–ECCV 2014: 13th European Conference, Zurich, Switzerland, September 6-12, 2014, Proceedings, Part VI 13. Springer International Publishing, 2014: 446-461.*

---

> > ### Comment · Reviewer_dVoG · 2023-11-21
> >
> > Thanks for your response.
> > In my understanding, given a predefined search space, there are two hyper-parameters for search in RandAugmentation: N (the number of augmentation transformations to apply sequentially) and M (magnitude for all the transformations). The core idea of the proposed SRA is adopting the Magnitude Instructor Score (MIS) to obtain M for each samples without search. However, the setting of N is still the same as RA.
> > Based on my understanding, I consider SRA is build upon RA since SRA needs the hyper-parameter N searched by RA, and the search space proposed by RA. In order to validate the generalization of the proposed MIS, I raise the following question in my first round review:
> > >What about adopting the proposed strategy with other augmentation methods? For example, randomly choosing augmentation operators from a search space, and comparing the performances with and without the proposed method.
> >
> > However, It seems that there is not direct response to this question. It would be helpful to address this point in order to validate the effectiveness of the proposed MIS strategy in comparison to other augmentation methods.
> >
> > Please let me know if I have any misunderstanding.

---

> > > ### Author Response · Authors · 2023-11-21
> > > **Response to dVoG**
> > >
> > > Thank you for your response. First we claim that SRA is inspired by the design of RA, but not totally following the design of RA. **The parameter $N$ for SRA is chosen as the empirical value by previous AutoDA rather than using the searched results from RA.** The parameter is denoted as depth $D$ in our paper. Note that we choose the configuration $D=2$ in all our experiments because it is a commonly chosen configuration in many previous AutoDA works, such as AA, FasterAA [1], DADA [2], etc.
> > >
> > > We have explored the impact of the parameter $N$ of RA, or $D$ in the table below, for SRA in the experiments in the Appendix Table A5. The configuration performs just well on CIFAR-100, which is consistent with RA. However, we also apply $D=2$ on CIFAR-10 using WRN-28-10 in the experiments in Table 1, while RA uses $N=3$ under the same setting. Our empirically selected configuration outperforms RA, which shows the advantages of SRA over its prototype.
> > >
> > > | $D$         | 1     | 2     | 3     | 4     |
> > > |-------------|-------|-------|-------|-------|
> > > | Accuracy (%)| 84.49 | **84.64** | 84.56 | 84.02 |
> > >
> > > In addition, it is possible to adopt SRA with other augmentation methods to yield better performance, while we believe current setting is simple and effective enough for general applications. **Since `Identity` is one of the operators in the search space of RA, the applied augmentation operators for SRA with $D$ has the probability to apply random number of operators, the number of which ranges from 0 to $D$.** As shown in the table, there is no linear relationship obsearved between performacne and the number of operators to apply, indicating the tuning of $D$ may further improve the performance of SRA. However, we believe keeping current configuration $D=2$ is just satisfactory for wide applications.
> > >
> > > We hope the response can solve your concerns.
> > >
> > > [1] Li Y, Hu G, Wang Y, et al. Differentiable automatic data augmentation[C]//Computer Vision–ECCV 2020: 16th European Conference, Glasgow, UK, August 23–28, 2020, Proceedings, Part XXII 16. Springer International Publishing, 2020: 580-595.
> > >
> > > [2] Hataya R, Zdenek J, Yoshizoe K, et al. Faster autoaugment: Learning augmentation strategies using backpropagation[C]//Computer Vision–ECCV 2020: 16th European Conference, Glasgow, UK, August 23–28, 2020, Proceedings, Part XXV 16. Springer International Publishing, 2020: 1-16.

---

> ### Author Response · Authors · 2023-11-16
> **Response to reviewer dVoG - Part 2**
>
> > **The proposed method highly depends on RandAugment. I consider the most important contribution of this paper to be the strategy of using MIS to adjust the magnitude of each sample. What about adopting the proposed strategy with other augmentation methods? For example, randomly choosing augmentation operators from a search space, and comparing the performances with and without the proposed method.**
>
> Thanks for the suggestion. The proposed MIS is the way to determine magnitude, thus it is possible to be integrated into other augmnetation frameworks that require magnitude selection. However, we are not optimistic to the results of integrating MIS to other augmentation methods, because we recognize our SRA as a whole augmentation design inspired by the simplicity of RA, rather than RA+MIS.
>
> We agree with the idea that the most important contribution of this paper is the strategy of using MIS to adjust the magnitude of each sample. The motivation behind this design is learning from hard samples that contain more information for deciding classification boundaries. However, learning from hard samples is no easy task, which may even bring negative effects to the representation ability of the target model. Fig. 6b shows the negative effects of over transformed hard samples, which form a small cluster in the center of the figure with samples from different classes.
>
> To reduce the negative effect of overfitting on hard samples that show a different data distribution to the original data, SRA adopts an asymmetric augmentation strategy that dynamically adjusts the represented data distribution during learning from hard samples. The random exploration step is important for SRA, as demonstrated in the ablation study (see Table 6, especially exp 3). We have also listed the results below for convenience. Therefore, ***SRA is not simply a combination of RA and MIS, but a whole augmentation strategy with MIS inspired by the design of RA.*** It is an alteration to RA, therefore can be orthogonally combined with other augmentation frameworks that require a data augmentation method as the basis, such as TiedAugment and BatchAugment. If integrating MIS alone into other data augmentation, the advances of MIS may not be fully ultilized.
>
> | Description                                                                      | CIFAR-10  | CIFAR-100 |
> | -------------------------------------------------------------------------------- | --------- | --------- |
> | 1) Remove $\gamma$                                                               | 97.60     | 84.49     |
> | 2) Remove random aug in Step 1                                                   | 97.47     | 83.93     |
> | 3) Replace Step 1 with Step 2 & 3 (Aug with MIS)      | 97.60     | 84.09     |
> | 4) Use Euclidean distrance for MIS                                               | 97.65     | 84.37     |
> | 5) Replace Step 2 & 3 with Step 1 (Aug with random mag) | 97.41     | 83.92     |
> | 6) Proposed SRA                                                                  | **97.67** | **84.64** |
>
> The ablation studies include the comparison of SRA and its variants. For example, exp 3 is the case where all the data are augmented using MIS. Exp 5 is the case where all the data select random augmentation operators and random magnitudes for augmentation. Although these variants also improve performance compared with RA, these settings are suboptimal compared with the asymmetric augmentation strategy. The results further demonstrate that SRA is not simply RA+MIS, but a whole augmentation design.

---

> ### Author Response · Authors · 2023-11-16
> **Response to reviewer dVoG - Part 3**
>
> > **What is the purpose of Step 1? Is it necessary to split the batch into two splits? (1) What about doing step 1 on the first batch and then doing steps 2 and 3 on the second batch? (2) What about removing step 1?**
>
> We've split this question into three parts.
>
> > **a. What is the purpose of Step 1?**
>
> The purpose of Step 1 is to dynamically adjust the represented data distribution after model inference to reduce the negative effect of overfitting on hard samples that show a different data distribution to the original data.
>
> > **b. Is it necessary to split the batch into two splits? (1) What about doing step 1 on the first batch and then doing steps 2 and 3 on the second batch?**
>
> It is not exactly required to split the batch into two splits. In most cases, the split strategy is identical to alternatively using Step 1 and Step 2 & 3  to train the model with twice the batchsize in traditional augmentation settings. However, when using epochs rather than iterations to count the training time, for the cases that the last batch only has samples less than one batchsize, only Step 1 will be applied while Step 2 & 3 are missed. To balance the number of exploration and refinement steps, we adopt a batch split strategy, and select batchsize twice of previous works in all our experiments for a relatively fair comparison. We guess the way doing step 1 on the first batch and then doing steps 2 and 3 on the second will show trivial effects to performance compared with the proposed batch split strategy.
>
> > **c. (2) What about removing step 1?**
>
> The removal of Step 1 has been studied in the ablation experiments (see Table 6 exp 2 and 3). Both the random augmentation in Step 1, and the exploration step itself, contribute to the satisfactory performance of SRA.

---

### Official Review · Reviewer_biM5 · 2023-11-01

**Soundness:** 2 fair
**Presentation:** 3 good
**Contribution:** 2 fair
**Rating:** 5
**Confidence:** 4

**Summary:**

This work proposes an asymmetric search-free augmentation strategy, named SRA, that can dynamically adjust the augmentation policy during the training procedure. Specifically, the authors split a batch into two sub-batches, one is applied with random data augmentation and the other is applied with a sample-aware data augmentation. First, the sub-batch is fed into the model concatenated by a MIS module, which will output the magnitude of the augmentation operators. Then the same sub-batch is augmented and fed into the model again to update the weights.

**Strengths:**

1.	The proposed method is straight-forward and easy to implement.

2.	Extensive experiments show the effectiveness of the proposed method.

**Weaknesses:**

1.	This method requires three times forward to update the weights twice, which can be inefficient.

2.	The augmentation operator in the sample-aware augmentation branch is fixed. I wonder how to design the augmentation operators since the selection will significantly affect the performance. The proposed MIS module simply outputs one scalar serving as magnitudes of the augmentation operators, leading to a quite small search space.

3.	I wonder if is there any theory supporting that the cosine similarity between logits and labels has a linear positive relationship with the magnitude of augmentation operators.

**Questions:**

Please see the weaknesses.

---

> ### Author Response · Authors · 2023-11-16
> **Response to Reviewer biM5**
>
> We sincerely appreciate your thoughtful comments, efforts, and time. We response to all your concerns as follows:
>
> > **This method requires three times forward to update the weights twice, which can be inefficient.**
>
> Thanks for your comment. We admit that the extra forward process increases the training time, but argue that it is also efficient in practical training. Since we only require the forward calculation results in Step 2 of SRA and doesn’t require recording the gradients, the Step 2 in practice is efficient. The real training cost, tested and reported in our paper, is also listed below. The extra time of SRA is <10%, which is generally as efficient as RA.
>
> | Model                   | WRN-28-10  | WRN-28-10  |
> | ----------------------- | ---------- | ---------- |
> | Dataset                 | CIFAR      | CIFAR      |
> | Aug Settings            | RA         | SRA        |
> | GPU                     | 1 RTX 3090 | 1 RTX 3090 |
> | Training Time (s/epoch) | 96         | 105        |
>
> In addition, the extra time can be further reduced. Reducing the frequency to calulate MIS can improve the efficiency. For example, training with Step 1 for several subbatches and Step 2 & 3 for one subbatch, is worth trying. Another way is introducing an auxiliary classifier (proposed in InceptionNet [1]) to the model, which can prune the calculation in deeper stages of the model during Step 2 and reduce the time to calculate logits.
>
> [1] *Szegedy C, Liu W, Jia Y, et al. Going deeper with convolutions[C]//Proceedings of the IEEE conference on computer vision and pattern recognition. 2015: 1-9.*
>
> > **The augmentation operator in the sample-aware augmentation branch is fixed. I wonder how to design the augmentation operators since the selection will significantly affect the performance. The proposed MIS module simply outputs one scalar serving as magnitudes of the augmentation operators, leading to a quite small search space.**
>
> Since our SRA is deeply inspired by the simple while effective RA, the selection of augmentation operators of SRA follows RA, which is not specifically designed. This search space is also competitive to many of other AutoDA methods, therefore allowing a relatively competitive or fair comparison. However, we also admit that the specific selection of augmentation operators are important for AutoDA, which may further improve the performance.
>
> To better select effective operators for SRA, we model the operator sampling process as a multi-armed bandit problem, and provide weights to operators (arms in the model) to indicate their importance. Therefore, the sampling process is seen as a one state optimization problem. The value of each operator is learnable, which is also set as the importance weight. We use the following equation to update weights:
>
> $Q(s,a) \gets (1 - \alpha) \cdot Q(s,a) + \alpha \cdot(r + \gamma \cdot \max_{a'} Q(s,a')),$
>
> where $s$ is the state, $Q(s,a)$ is the weight for operator $a$, $r$ is the reward evaluated using the average predicted probability of the correct class for samples augmented using $a$, $\alpha$ is the learning rate for updating weights, and $\gamma$ is the discount factor. The learned weights are dynamically applied in Gumbel-Softmax sampling [1] for operator selection.
>
> | Model               | WRN-28-10      | WRN-28-10      |
> | ------------------- | -------------- | -------------- |
> | Dataste             | CIFAR-10       | CIFAR-10       |
> | Aug Settings        | SRA            | SRA+$OP_w$     |
> | Final Training Loss | 0.1543         | 0.1500         |
> | Accuracy (%)        | 97.67$\pm$0.02 | 97.70$\pm$0.09 |
>
> As shown, the performance of SRA slightly increases with the application of operator sampling. Changing the optimization method may further improve the performance. However, the original intention of our SRA is proposing a simple while effective augmentation method for wide application. Although introducing the learnable sampling weight of operators may improve the performance, it also increases the complexity of this  method. Therefore, we prefer the simple random sampling strategy, which is just satisfactory and straightforward for understanding and implementation in image recognition tasks. The small search space is the balanced result between performance gain and simplicity.
>
> [1] *Jang E, Gu S, Poole B. Categorical reparameterization with gumbel-softmax[J]. arXiv preprint arXiv:1611.01144, 2016.*

---

> ### Author Response · Authors · 2023-11-16
> **Response to Reviewer biM5 - Part 2**
>
> > **I wonder if is there any theory supporting that the cosine similarity between logits and labels has a linear positive relationship with the magnitude of augmentation operators.**
>
> Thanks for your comments. In fact, the cosine similarity between logits and labels doesn’t have a linear positive relationship with the magnitude of augmentation operators. We analyze the reason MIS module works is that the positive effect of more information from hard samples for determining accurate decision boundaries overwhelms the negative effect of learning on hard samples or even outliers.
>
> Images augmented with larger magnitudes are expected to deviate more from its original position in the data space, which increases the possibility to generate hard samples. However, learning from hard samples is no easy task, therefore the contributions of hard samples are not always positive to model learning. This observation is also shown in Fig. 3b in our paper, as the small cluster in the center of the figure is constructed by a set of SRA augmented samples from different classes.
>
> Cosine similarity between logits and labels is an intuitive way to measure the difference of the predicted result to the label, which reflects the difficulty to learn from the samples. Therefore, cosine similarity used in SRA achieves generating more hard samples. To reduce the risk on overfitting on the hard samples, SRA also adopts an asymmetric augmentation strategy to dynamically adjust the represented data distribution during training. Therefore, SRA strikes the balance between learning from hard samples and be degraded due to too many hard samples, which results in a positive effect of current MIS that uses cosine similarity between logits and labels as the magnitude of augmentation operators.
>
> The analysis of why SRA works have been modified in the paper as well.

---

### Official Review · Reviewer_uXgv · 2023-11-01

**Soundness:** 2 fair
**Presentation:** 2 fair
**Contribution:** 2 fair
**Rating:** 3
**Confidence:** 3

**Summary:**

This work proposes a search-free sample-aware automatic data augementation strategies, in which a heuristic metric is proposed to evaluate the difficulty of  training samples. Such evaluation results are used to guide the generation of augmented samples that contribute to decision boundaries during training. Further, an asymmetric data augmentation strategy is proposed.

**Strengths:**

The research question this work focuses on holds significant research value. It is meaningful to study how to perform data-aware augementation instead of augmenting with an entirely random strategy.

**Weaknesses:**

The biggest problem is the targeted issue has been well-studied in SelectAugment [1], which is also a sample-aware data augmentation strategy learned using RL. Compared to the heuristic design, RL-based learning might be more generally applicable. The authors omit this for necessary comparison and analysis. This also makes the technical contributions unclear and makes the scope of contribution scope be quite limited.

[1] Lin, Shiqi, et al. "SelectAugment: hierarchical deterministic sample selection for data augmentation." Proceedings of the AAAI Conference on Artificial Intelligence. Vol. 37. No. 2. 2023.

**Questions:**

Are the proposed method sensitive to the configurations of data augmentation operators?

What is the scope of application for the proposed method? Are there any limitations in its use?

---

> ### Author Response · Authors · 2023-11-16
> **Response to Reviewer uXgv**
>
> We sincerely appreciate your thoughtful comments, efforts, and time. We response to all your concerns as follows:
>
> > **The biggest problem is the targeted issue has been well-studied in SelectAugment [1], which is also a sample-aware data augmentation strategy learned using RL. Compared to the heuristic design, RL-based learning might be more generally applicable. The authors omit this for necessary comparison and analysis. This also makes the contributions unclear and makes the scope of contribution scope be quite limited.**
> >
> > **[1] Lin, Shiqi, et al. "SelectAugment: hierarchical deterministic sample selection for data augmentation." Proceedings of the AAAI Conference on Artificial Intelligence. Vol. 37. No. 2. 2023.**
>
> Thanks for pointing out the previous related work that is missed from our related work.  Here, we claim and emphasize that the purpose of the sample-aware design between SelectAugment and SRA are different.
>
> ##### **(1) We emphasize that the sample-awareness of our SRA affects the magnitude of augmentation operators, while that of SelectAugment affects the selection of whether to apply augmentation to each input sample.**
>
> Although both SRA and SelectAugment adopt sample-aware strategies to achieve effective augmentation, the purposes are different. For SelectAugment, it provides scores to determine whether to apply augmentation to the original data. On the contrary, SRA provides scores to determine the deformation level of the augmentation to the original data. In other words, ***SelectAugment determines whether to do augmentation, while SRA determines how to do augmentation.***
>
> ##### **(2) The focus of SRA is to propose a simple while effective AutoDA method, rather than studying the effectiveness of sample-awareness.**
>
> The target issue in SRA is how to achieve both satisfactory performance and simplicity for AutoDA. Previous works, such as MetaAugment mentioned in the Related Work, and SelectAugment mentioned here, have shown the importance of sample-awareness to achieve effective augmentation. Therefore, our work directly adopts the results found in these works, which proposes a new augmentation method to be sample-aware. ***Sample-awareness is a means of implementation to achieve the effective augmentation, rather than the target issue to study in SRA.***
>
> ##### **(3) We argue that the heuristic design is simpler and more straightforward for general application than RL-based optimized strategy.**
>
> We agree with the idea that RL-based methods show higher upper bounds in general tasks. However, RL-based methods usually require expert knowledge and careful hyperparameter tuning to achieve stable and good performance. On thecontrary, our heuristic design is simple and straightforward, which almost requires no hyperparameter tuning to achieve a satisfactory performance. The search-free design is expected to significantly reduce the time consumption to adopt the augmentation method to new tasks. Therefore, we argue that the heuristic design is easier for general application.
>
> ##### **(4) We argue the contributions and the contribution scopes of our SRA is valuable to the community.**
>
> We have added comparison and analysis of SelectAugment in Related Work. We emphasize that the main technical contributions of SRA as two parts.
>
> 1) Proposing a MIS module for scoring the difficulty of the original data during training to dynamically adjust the augmentation magnitudes in a sample-aware manner, which demonstrates that search-free heuristic design can also achieve satisfactory performance compared with search-based ones. This finding may raise future works to focus more on simple, effective, and practical augmentation designs. Meanwhile, it also provides a new insight of sample-awareness to focus on the augmentation operators for samples, rather than the importance of samples.
>
> 2) Proposing an asymmetric augmentation strategy that is different from previous augmentation methods to better ultilize the designed MIS module. It provides a new train of thought that the design of hybrid data augmentation is also important to fully release the power of data augmentation in neural network training. More hybrid augmentation designs are worth exploring, which may boost the finetuning process of current focus large language models in downstream tasks in the future.
>
> Thanks again for your comments on our work. We have modified related parts in the paper to clearly state the points above. We are looking forward to your further discussions.

---

> ### Author Response · Authors · 2023-11-16
> **Response to Reviewer uXgv0 - Part 2**
>
> > **Are the proposed method sensitive to the configurations of data augmentation operators?**
>
> Yes, just like other AutoDA methods, SRA is sensitive to augmentation operators. We conduct operator ablation experiments on CIFAR-10 using WRN-28-10 to show the sensitivity of SRA in the following and Appendix. For each line we delete one specific operator from the original 14 operators, and report the performance of SRA on the remaining 13 operators.
>
> | Ablation Operator | Accuracy (%)   | Operator Gain                | Ablation Operator | Accuracy (%)   | Operator Gain                |
> | ----------------- | -------------- | ---------------------------- | ----------------- | -------------- | ---------------------------- |
> | ShearX            | 97.78$\pm$0.02 | -0.11| Sharpness         | 97.58$\pm$0.03 | 0.09                         |
> | ShearY            | 97.56$\pm$0.05 | 0.11                         | Contrast          | 97.65$\pm$0.06 | 0.02                         |
> | TranslateX        | 97.70$\pm$0.01 | -0.03 | Solarize          | 97.60$\pm$0.11 | 0.07                         |
> | TranslateY        | 97.60$\pm$0.07 | 0.07                         | Posterize         | 97.51$\pm$0.05 | 0.16                         |
> | Rotate            | 97.58$\pm$0.05 | 0.09                         | Equalize          | 97.74$\pm$0.01 | -0.07 |
> | Brightness        | 97.74$\pm$0.06 | -0.07 | Autocontrast      | 97.59$\pm$0.02 | 0.08                         |
> | Color             | 97.59$\pm$0.08 | 0.08                         | Identity          | 97.61$\pm$0.04 | 0.06                         |
> | None (SRA)        | 97.67          | -                            |                   |                |                              |
>
> As shown, the removal of ShearX, TranslateX, Brightness, and Equalize from the configurations of SRA data augmentation operators may further improve the performance. However, we simply follow the search space of RA, which is also the widely applied AutoDA method, for a relatively fair comparison. The the combination of SRA with RL or gradient optimization strategy is also worth trying to dynamically select the important and positive augmentation operators during training, which may further boost the performance. However, the original intention of our SRA is a simple while effective method for wide applications, therefore we simply choose the search space of RA.
>
> > **What is the scope of application for the proposed method? Are there any limitations in its use?**
>
> Currently, our SRA focuses on the supervised tasks for image recognition. Since the proposed MIS module relies on the logits to calculate the scores that indicate the difficulty of the data in the batch, it requires labels that are generally provided in supervised tasks. However, the most important design of SRA is adopting a sample-aware scoring module in the current augmentation pipeline, rather than using a specific scoring method. Therefore, with modifications to the MIS formulation that are agnostic to labels, the proposed SRA can be adopted into unsupervised and self-supervised tasks.
>
> Besides, current SRA explores the effectiveness on image recognition tasks. With adjustments to augmentation operators like bounding box augmentation, the effectiveness of SRA in downstream tasks, such as object detection, is worth trying and analyzing, which is also our future work.

---

### Author Response · Authors · 2023-11-16
**Summary of the Rebuttal**

We deeply appreciate the efforts AC and all three reviewers made to our submission in ICLR 2024. We propose a search-free sample-aware automatic data augmentation method SRA that is expected to be easily integrated into image recognition tasks, with general improvements to the widely used automatic data augmentation method RandAugment (RA). The comments on our work SRA are valuable for us to further improve our work, which are worthwhile for further discussions as well. We have revised our manuscrip in the following aspects, and response to all reviewers points-by-point.

### Discussions:

- Add a missed sample-aware AutoDA method SelectAugment to Related Work, and analyze the relation and differences of it compared with our SRA. Most importantly, we claim that ***SelectAugment determines whether to do augmentation or not, while SRA determines how to do augmentation.*** (Reviewer uXgv)

- Discuss about the sensitivity of SRA to different augmentation operators and the potential of better utilizing augmentation operators through expanding the search space. (Reviewer uXgv, biM5)

- Discuss about the scope of application for the proposed SRA and limitations in its use. (Reviewer uXgv)

- Emphasize the efficiency of SRA in practice, and provide applicable ways to further improve its efficiency. (Reviewer biM5)

- Analyze why SRA works with the cosine similarity between logits and labels to determine the magnitude between logits and labels. (Reviewer biM5)

- Further discuss about the relation between SRA and RA, demonstrate the generalization advantages of SRA in new tasks, and emphasize the design of SRA as an alternation to RA rather than MIS+RA. (Reviewer dVoG)

- Discuss about the necessity of asymmetric augmentation design in SRA. (Reviewer dVoG)

### Experiments:

- Add an ablation experiment to explore the impact of each augmentation operator, which shows the sensitivity of SRA to different augmentation operators.

- Explore SRA using a larger search space, with learnable sampling weights to select augmentation operators rather than random sampling.

- Add a new image recognition benchmark Food101 [1] , which is a more fine-grained dataset compared with ImageNet, to demonstrate the generalization advantages of SRA.

- Note: Newly added experiments are run for 3 times to evaluate the performance.

- Emphasize the ablation study in Table 6 to show the necessity of asymmetric augmentation design in SRA.

Detailed responses are replied to each reviewer. Thanks again for your discussion. We hope the response can address all your concern. We are willing to further disscuss with reviewers, AC, and anyone who is interested in our SRA.

---

### Author Response · Authors · 2023-11-20
**Looking Forward to Further Feedback**

Dear Reviewers,

Thank you again for your valuable comments and suggestions, which are really helpful for us. We have posted responses to the proposed concerns.

We totally understand that this is quite a busy period, so we deeply appreciate it if you could take some time to return further feedback on whether our responses solve your concerns. If there are any other comments, we will try our best to address them.

Best,

The Authors of Submission72

---

### Meta-Review · Area_Chair_FLZF · 2023-12-11

**Metareview:**

In this paper the authors propose a new form of data augmentation for training deep neural networks. In particular, the authors propose a data augmentation policy that dynamically updates during the course of training. The authors test their method on image classification mostly through experimentation on the ResNet architecture on CIFAR-10/100 and ImageNet. The authors also show some experiments using the vision transformer model (via DeiT) on ImageNet image classification. The reviewers commented positively on the relative simplicity of the method and the extensive experiments. The reviewers also commented negatively on the marginal gains exhibited by the method. The authors provided some detailed rebuttals arguing against these concerns, however none of the reviewers responded to these comments. For these reasons, the AC took a closer look at the paper to see if I may be able to argue for acceptance. Upon my reading of the paper, I shared the reviewers' concerns about the marginal gains in improvement on CIFAR-10/100 and ImageNet. Additionally, most of the experiments focused on training with ResNet with CIFAR-10 and thus the general applicability of these results are limited. For all of these reasons, the AC can not argue for acceptance and this paper will not be accepted to this conference.

**Justification For Why Not Higher Score:**

Marginal gains. Limited experiments.

**Justification For Why Not Lower Score:**

N/A

---

### Decision · Program_Chairs · 2024-01-16

Reject